# Molecular Basis of Pancreatic Neuroendocrine Tumors

**DOI:** 10.3390/ijms252011017

**Published:** 2024-10-14

**Authors:** Alesia Maluchenko, Denis Maksimov, Zoia Antysheva, Julia Krupinova, Ekaterina Avsievich, Olga Glazova, Natalia Bodunova, Nikolay Karnaukhov, Ilia Feidorov, Diana Salimgereeva, Mark Voloshin, Pavel Volchkov

**Affiliations:** 1Moscow Center for Advanced Studies, Kulakova Str. 20, Moscow 123592, Russia; alesyamaluchenko@gmail.com (A.M.); maksimov.denis.o@gmail.com (D.M.); zoya.antysheva@gmail.com (Z.A.); e.avsievich@mknc.ru (E.A.); ol.glazova@gmail.com (O.G.); vpwwww@gmail.com (P.V.); 2Moscow Clinical Scientific Center N.A. A.S. Loginov, Moscow 111123, Russia; n.bodunova@mknc.ru (N.B.); n.karnaukhov@mknc.ru (N.K.); i.feidorov@mknc.ru (I.F.); d.salimgereeva@mknc.ru (D.S.); m.v.voloshin@mknc.ru (M.V.)

**Keywords:** pancreatic neuroendocrine tumors, molecular classification, prognosis, omics technologies

## Abstract

Pancreatic neuroendocrine tumors (NETs) are rare well-differentiated neoplasms with limited therapeutic options and unknown cells of origin. The current classification of pancreatic neuroendocrine tumors is based on proliferative grading, and guides therapeutic strategies, however, tumors within grades exhibit profound heterogeneity in clinical manifestation and outcome. Manifold studies have highlighted intra-patient differences in tumors at the genetic and transcriptomic levels. Molecular classification might become an alternative or complementary basis for treatment decisions and reflect tumor biology, actionable cellular processes. Here, we provide a comprehensive review of genomic, transcriptomic, proteomic and epigenomic studies of pancreatic NETs to elucidate patterns shared between proposed subtypes that could form a foundation for new classification. We denote four NET subtypes with distinct molecular features, which were consistently reproduced using various omics technologies.

## 1. Introduction

The incidence of pancreatic neuroendocrine tumors (NETs) is 0.48 per 100,000 persons, according to the US-based SEER 18 database, 0.33 in Iceland, 0.94 in Norway, and 2.2 in Japan [1]. Pancreatic NETs are less common than those in the lungs, rectum and small intestine [2]. Their development is assumed to be affected by several environmental and lifestyle influences. According to systematic reviews and meta-analyses, they include family history of malignancy, history of diabetes, tobacco smoking and heavy alcohol consumption [3,4]. NETs of pancreatic localization are estimated to have the worst median overall survival, of 3.6 years, among site groups [2]. Like most NETs of other localizations, they express chromogranin A, synaptophysin, and neuron-specific enolase. In contrast to other types of NETs, pancreatic NETs are highly heterogeneous in clinical manifestation. Approximately 30% of patients are diagnosed at a metastatic stage, due to an asymptomatic course of disease [5]. Although the majority of specimens exhibit no hormone hypersecretion and are referred to as non-functional NETs, functional tumors include insulinomas, gastrinomas, glucagonomas, somatostatinomas, VIPomas, and other less common types with pronounced clinical manifestations, due to hormone excess. Hormonal syndromes that aid in diagnosis are a unique feature of gastroenteropancreatic NETs. The variability in malignancy [6] and hormone production suggests the distinct origin and pathogenesis of each tumor type. The existence of gastrinomas, for instance, calls into question the hypothesis that all NETs develop from hormonally programmed precursor cells [7], as distinct gastrin-positive cells were not found in the adult human pancreas. Conversely, a number of mouse studies have indicated that islet cells exhibit plasticity and transdifferentiation, even from the exocrine component of the gland [8,9,10]. Furthermore, the deactivation of *Men1* gene in murine glucagon-secreting alpha cells may result in the development of an insulin-hypersecreting tumor, whereas the deactivation of this gene in insulin-secreting beta-cells may result in glucagonoma [11,12,13]. The cancer stem niche is another potential source of neoplastic cells [14]. There are conflicting opinions about mechanisms of self-renewal in the endocrine pancreas. A mice study conducted by Dor et al. suggest that beta cells regenerate through self-duplication, although slowly [15]. In contrast, several studies showed that ducts of extrahepatic biliary trees contain stem cells and committed progenitors. The hepato-pancreatic common duct and pancreatic ducts may represent the source of pancreatic endocrine progenitors, expressing markers of early hepato-pancreatic commitment (SOX9, SOX17, PDX1), as well as markers of pancreatic endocrine maturation (NGN3 and insulin), and which are capable of differentiating into endocrine cells in vitro and in vivo [16,17,18]. Therefore, the origin of neuroendocrine tumors remains obscure.

According to the current edition of the World Health Organization Classification of Endocrine and Neuroendocrine Tumors (WHO 2022), pancreatic neuroendocrine neoplasms are divided into two distinct groups: neuroendocrine tumors (NETs) and neuroendocrine carcinomas (NECs) [19]. These two groups exhibit cytohistological differences and unique immunohistochemistry profiles. Neuroendocrine tumors represent the group with preserved morphological features. Their further classification is based on proliferative grading (grade 1, grade 2, and grade 3) and is rated by mitotic rate and Ki67 staining index. At the same time, neuroendocrine carcinomas are poorly-differentiated, high grade, by definition, and are subclassified as small-cell and large-cell NECs. Here, we will investigate the neuroendocrine tumor biology, as neuroendocrine carcinomas are considered to have remarkable dissimilarities related to both cytohistological and genetic changes, along with pathogenesis and clinical management, even when compared to high-proliferating tumors [20,21,22,23]. Mutations in TP53 and RB, which are essential early drivers in pancreatic NECs but not NETs [24], and the expression of exocrine lineage markers, support the hypothesis of a unique entity of NECs, and the closer similarity of NECs to ductal adenocarcinomas than to grade 3 NETs [20]. However, neuroendocrine neoplasms should not be confused with pancreatic ductal adenocarcinomas, and can be distinguished from them primarily by neuroendocrine cell morphology, diffuse chromogranin A, synaptophysin, and neuron-specific enolase staining.

As new knowledge about NETs diversity accumulates, their classification is systematically reviewed and improved, although minor imperfections persist [25]. Nevertheless, the WHO 2022 does not address tumor intrinsic features that guide tumor progression. The use of genetic profiling technologies has revealed significant inter-patient heterogeneity between patients within the same WHO subtype during pancreatic neuroendocrine tumor research. High molecular heterogeneity is observed in grade 1 and 2 groups.

Molecular classifications of neuroendocrine tumors have already been introduced in clinical practice. Pituitary neuroendocrine tumor groups are defined based on hormone staining, along with transcription factor staining, which can infer terminally differentiated or immature cell types. Pituitary NETs classification is proposed to serve as a model for NETs in other primary sites [26].

It would appear that there is a finite number of classes that can encompass possible outcomes. While complementary molecular–genetic classification might not reveal the precise tumor origin, it can be aimed at reflecting cellular processes and features of tumor tissue that will stratify patients and guide therapeutic strategies. However, a precision approach for each individual, if available, is preferred. Nevertheless, the classification problem is complicated by limited therapeutic opportunities and the rarity of pancreatic neuroendocrine neoplasms. At present, surgical resection represents the sole potentially curative intervention for pancreatic NETs. There is currently no established standard of care for advanced disease. Available treatment options include the use of somatostatin analogues, peptide receptor radionuclide therapy, everolimus, sunitinib, and alkylating-based chemotherapy [27,28]. The development of new therapeutics is ongoing, and new models to facilitate this process are being developed. For example, a recently presented mouse model has been shown to accurately recapitulate pancreatic NETs, although it failed to form metastasis. It may, therefore, play a decisive role in elucidating the mechanisms of tumor progression and facilitate the development of effective therapeutic strategies [29]. As for now, there are a limited number of ways to impact therapeutic strategy. Identifying patients with higher risk of metastasis and recurrence may influence the decision on radical resection, chemotherapy use and frequency of postoperative monitoring. In addition, defining groups of tumors with common molecular and/or genetic alterations may aid in the development of novel group-specific therapies and in conducting clinical trials on narrow groups of people.

A multitude of studies have illustrated the complexity and ubiquity of alterations in neoplasms, highlighting the need for multidimensional analysis of the pathology. Currently available approaches to investigate the origin of a tumor and associate its characteristics with the outcome and benefit from existing therapeutic strategies comprise, but are not limited to, omics profiling.

In this review, we summarize the published studies that utilized genome, transcriptome, proteome and epigenome data to classify pancreatic neuroendocrine tumors (Figure 1). Overall, the results of many studies were found to be consistent, although some discrepancies were observed, which were mainly due to the limited sample size. Therefore, the feasibility of the classifications should be further evaluated on large cohorts. Additionally, new strategies that might aid in dissecting the challenging and strikingly diverse biology of NETs are discussed.

## 2. Genetic Patterns in NETs

### 2.1. Familial Syndromes

The genome landscape of pancreatic well-differentiated neuroendocrine neoplasms was comprehensively investigated on large cohorts, although some pathogenic alterations were identified in case studies. The majority of NETs are sporadic in nature, representing approximately 85–90% of cases [30,31]. However, they can also arise in familial syndromes, typically in an autosomal-dominant pattern.

Approximately 25–45% of tumors exhibit either sporadic or inherited alterations in the MEN1 gene (Multiple endocrine neoplasia type 1, encoding menin), with prevalence in non-functioning tumors [32,33]. Patients diagnosed with MEN1 syndrome are predisposed to develop tumors in the pituitary, parathyroid, pancreas, and other tissues. Menin is a nuclear protein that regulates transcription of multiple genes and interacts with various signaling pathways to affect proliferation, telomerase activity, and homologous recombination-directed DNA repair [34,35]. The inactivation of MEN1 is often biallelic (in 40% of cases [30]) and somatic or germline point mutations in the gene are coupled with loss of heterozygosity (LoH) at the MEN1 alleles in tumor tissue. Consequently, mutations in MEN1 have a multifaceted impact on a wide range of essential cellular processes, thereby highlighting the tumor suppressor role of the protein.

In addition to MEN1 syndrome, other hereditary diseases, such as von Hippel–Lindau, neurofibromatosis type 1, tuberous sclerosis complex, and Cowden syndrome have been associated with the development of various NETs [30,36,37]. Only a subset of patients with NETs carry mutations in VHL—a negative regulator of hypoxia signaling. VHL-associated tumors are characterized by small vessels, stromal collagen bands, and clear-cell morphology, with a predisposition to arise in the central nervous system, retina, kidney, and adrenal gland. Tumors with altered VHL function have been observed to be enriched with pseudohypoxia-associated RNAs [38]. Recently, Lee et al. [39] reported a neuroendocrine microtumor with a nodule-in-nodule pattern in a patient with VHL syndrome. The two nodules exhibited striking disparities in Ki-67 labeling indices, with 2.1% for the outer and 44.3% for the inner nodules. It is noteworthy that the HIF-2α inhibitor belzutifan represents a novel promising non-interventional option for inherited VHL-related pancreatic NETs, and has already been FDA-approved for renal cell carcinoma [40].

Neurofibromatosis Type 1 (NF1) is one of the most prevalent genetic disorders. Patients with NF1 disorder typically exhibit abnormalities in neural crest-derived tissues [37], although, on rare occasions, they develop pancreatic NETs [41]. Another subset of NET patients are genetically predisposed to mammalian/mechanistic target of rapamycin (mTOR) pathway dysregulation, due to suffering from tuberous sclerosis complex (affecting TSC1, TSC2) [42] or Cowden syndrome (affecting PTEN) [43].

Germline CDKN1B mutations are rarely observed in patients clinically suspected of MEN1 mutations, and are referred to as multiple endocrine neoplasia type 4 syndrome [30,44]. CDKN1B is the gene coding for the cyclin-dependent kinase inhibitor p27 and is nominated as a tumor suppressor. DNA damage repair genes, namely the base-excision-repair MUTYH, homologous recombination genes CHEK2 and BRCA2, also rarely contain germline pathogenic variants [30]. Glucagon cell hyperplasia and neoplasia, or Mahvash disease [45] (the only autosomal-recessive disorder found in NETs), insulinomatosis [46], succinate dehydrogenase (SDHx)-associated disease involving SDHD gene [47], and Lynch syndrome [48] are of particular rarity among familial syndromes of pancreatic NETs.

### 2.2. Sporadic Forms

Sporadic forms of the disease are most commonly driven by somatic mutations in the MEN1 (45%), DAXX (25%), and ATRX (20%) genes [32]. The two latter genes experience two-hit inactivation in 20% of well-differentiated neoplasms analogous to MEN1. The death domain-associated protein (Daxx) and α-thalassemia X-linked mental retardation protein (ATRX) form a complex that enables the incorporation of the H3.3 histone variant at telomeres [49]. ATRX targets G/C-rich tandem repeats that are frequently found at telomeric and subtelomeric regions, and is likely to interact with G-quadruplex structures [50]. A number of studies have demonstrated a strong association between DAXX/ATRX mutational status and increased telomere length [30,38,51,52,53,54]. The vast majority of tumors exhibiting DAXX/ATRX losses exhibit an alternative lengthening of telomeres (ALT)—telomerase-independent mechanism of telomere maintenance, which can be examined using affordable C-tailing qPCR or telomere-specific fluorescence in situ-hybridization methods. It is noteworthy that the ALT phenotype is associated with poor prognosis [55]. Furthermore, chromosome instability, which is typically quantified as the number of gains and losses, is another common pattern observed in DAXX/ATRX-mutant specimens [52]. Well-differentiated NETs belonging to this group are associated with larger tumor size, higher grade, the presence of lymph node metastasis and distant metastasis, and generally demonstrate an unfavorable prognosis [52,53].

Transcription of the phosphatidylinositol 3-kinases (PI3K), protein kinase B (AKT) and the mammalian/mechanistic target of rapamycin (mTOR) pathway genes and its regulators is comprehensively dysregulated in pancreatic NETs [56]. Firstly, somatic mutations are found in PIK3CA (encoding a catalytic subunit of phosphatidylinositol 3-kinase), negative regulators of the mTOR pathway (PTEN, TSC2, TSC1), DEPDC5 encoding a subunit of an mTOR suppressor complex, and FLCN encoding a positive modulator of mTORC1 activity [30,32,38,57]. Mutations in these genes result in the activation of the PI3K/Akt/mTOR signaling pathway. Secondly, PI3K can be dysregulated by gene copy-number variations, with the most common being amplification in PSPN (PSPN binds RET and activates PIK3CA). Interestingly, Scarpa et al. [30] identified the involvement of the EWSR1 gene in fusion with BEND2 and FLI1 and hypothesized the role of the chimeric genes in mTOR signaling upregulation. Furthermore, one sample from the cohort exhibited an EWSR1 exon 7–FLI1 exon 5 fusion, and demonstrated positive staining for CD99, a marker characteristic of sarcomas. However, these tumors are extremely rare in the pancreas. Despite the prevalence of mTOR dysregulations, not all of them can be targeted for therapeutic intervention. Nevertheless, up to 15% [32] of all NET patients (i.e., the percentage of patients with mTOR dysregulations) may potentially benefit from the inhibitory therapy. Furthermore, patients with progressive advanced pancreatic NETs who were treated with the mTOR inhibitor, everolimus, exhibited prolonged progression-free survival [58].

Another gene involved in tumor progression is YY1, which encodes the transcription factor Yin Yang 1. In pancreatic beta cells, YY1 affects insulin production, glucose tolerance, mitochondrial function, proliferation, apoptosis, cellular senescence, and DNA damage recognition and repair [59,60,61]. YY1 mutations do not occur in non-functioning NETs [57], but a somatic gain-of-function T372R substitution is found in approximately one-third of insulinomas [62], resulting in an adenylyl cyclase and calcium channel increased expression and the subsequent dysregulation of insulin secretion [63].

Structural rearrangements and aneuploidy, which may affect hundreds of genes, are reported to be frequent in pancreatic NETs. Structural rearrangements may result in inactivation of MTAP, ARID2, SMARCA4, MLL3, CDKN2A, and SETD2, which are widely-recognized tumor suppressor genes. Recurrent losses are present in MEN1, CDKN2A, EYA1, FMBT1, and RABGAP1L genes, whereas regions of amplification include PSPN and ULK1 [30].

Lawrence et al. [38] demonstrated that recurrent aneuploidy is also present in primary NETs. In the study, 77% of specimens exhibited loss of a copy of ≥1 chromosome, while 79% exhibited loss of heterozygosity of ≥1 chromosome, and 26% exhibited loss of heterozygosity of ≥8 chromosomes.

### 2.3. Genetic Subtypes

A number of studies have been conducted with the aim of identifying consistent patterns in genome alterations in pancreatic NETs and linking them to potential therapeutic avenues and prognosis.

Lawrence et al. [38] analyzed 57 sporadic pancreatic NETs through the use of deep hybridization-capture sequencing of 637 genes and RNA sequencing (RNA-seq), revealing three groups of NETs. One group was characterized by a recurrent LoH pattern in 10 chromosomes (1, 2, 3, 6, 8, 10, 11, 16, 21, and 22). Additionally, 9 out of 10 patients in this group were MEN1-mutant. Additional alterations included DAXX, ATRX, PTEN, and TP53. The group comprised the only three patients who progressed during the study, and all but one patient presented with lymphovascular invasion. Metastasis and a high grade were significantly associated with the group.

The second group (16 patients) demonstrated frequent LoH in the 11th chromosome, MEN1 mutations, the absence of metastasis, and the rarity of lymphovascular invasion. The majority of tumors had grade-1 and expressed GCG.

Consequently, the two groups with distinct LoH patterns exhibited different clinical outcomes. Moreover, the authors suggested aneuploidy to be a plausible major driver of tumor progression in these subsets of specimens.

The remaining group exhibited a variable clinical outcome and lack of MEN1 mutations. The group might include other less frequent subtypes, necessitating an augmentation of the cohort size to investigate their genomic and molecular features.

In numerous studies, specimens harboring MEN1/DAXX/ATRX mutations were confirmed to constitute a group with unique genomic, transcriptomic and epigenomic features. For instance, Chan et al. [64] divided the cohort of 64 G1/G2 patients into two groups: those with ATRX/DAXX/MEN1 mutations in the primary specimens and those without these mutations. They then compared the RNA level and methylation profiles between the two groups. It is noteworthy that unsupervised clustering of expression or DNA methylation data revealed the same two groups, which demonstrated profound differences at the molecular and epigenetic levels. ATRX/DAXX/MEN1 wild-type tumors exhibited high heterogeneity, casting doubt on their ability to represent a single subset of NETs. In contrast, the expression profiles of the other group were found to be correlated, with specimens exhibiting less variance in principal component space.

In Chan’s research, the expression of different adult pancreatic cell-type signatures demonstrated a transcriptomic resemblance of mutant tumors to alpha cells. Conversely, while some alpha cell marker genes, ARX, IRX2, and TM4SF4, were enriched in mutant specimens, GCG (encoding the precursor of glucagon, the main hormone produced by alpha cells) was not. A subset of tumors from the wild-type cohort exhibited elevated expression of beta cell-specific genes. However, the signature was not significantly enriched in this group as a whole.

The search for activated molecular pathways revealed that the complement and coagulation cascades, retinol metabolism, steroid hormone biosynthesis and drug metabolism were up-regulated in mutant NETs. The high and differential expression of HNF1A and its regulator HNF4A, as well as genes having HNF1A transcription factor motifs, suggests the involvement of liver-associated processes. Additionally, genes involved in protein secretion, transport, and metabolism pathway genes (APOH, ALB, AFM, HAO1, UGT1A3, UGT1A1, GC, G6PC, TM4SF4, PKLR, etc.) exhibited high and differential expression in the mutant group. Notably, many of these genes are targets of HNF1A. The authors observed that APOH could serve as a marker of ATRX/DAXX/MEN1-mutant NETs, as its expression was more than eightfold higher, and approximately 70% of specimens were APOH-positive, compared to 18% in the wild-type group. It is noteworthy that the PDX1 gene appeared to be hypermethylated and repressed in the mutant group, while the ARX gene was not among the differentially methylated (which will be discussed in detail in the Section 4). Finally, mutant tumors exhibited worse recurrence-free survival, larger size, and higher stage.

In both studies, a considerable number of tumors were non-functional, and only isolated cases of hormone-producing ones were included in the analysis. The genetic landscape of functioning tumors is scarcely investigated, because of their rarity. Thus, Hong et al. [57] focused on differences between non-functioning NETs and insulinomas. The authors analyzed WGS/WES data of 84 insulinomas and 127 non-functional NETs. Insulin-producing tumors exhibited smaller size, lower grade, and lower tumor-mutation burden, consistent with their generally more beneficial prognosis. The results of somatic CNV inferring were subjected to unsupervised clustering to combine samples into five groups (Ins-Amp, Ins-Neutral, NF-Amp, NF-Neutral, and NF-Del), named in accordance with CNV patterns (amplification, deletion, and copy-neutral).

The two insulinoma subtypes (Neutral and Amp) demonstrated no significant differences in relapse-free survival (both had several relapses), grade and size. Chromosome 11 loss was extremely rare in insulinomas, and three out of four such specimens fell within the copy-neutral subtype. mTOR pathway activity (as measured by immunohistochemistry staining for p-S6) appeared to be significantly higher only in copy-neutral insulinomas, independent of YY1 mutation. However, mTOR pathway-related genes were frequently amplified in another insulinoma subtype. The YY1 gene mutation was frequent only in copy-neutral insulinomas (Ins-Neutral). This suggests that the two insulinoma subtypes may represent different developmental routes and molecular alterations, yet demonstrate similar clinical outcomes.

Amplification events were observed on different chromosomes in the two hormonal status groups. The early amplifications were observed in chromosomes 3, 5, 7, and 13 in insulinomas, and in chromosomes 4 and 14 in non-functional tumors. Furthermore, insulinomas with amplification patterns exhibited no recurrent mutations.

Non-functional tumors with deletions (NF-Del) exhibited the highest tumor-mutation burden. The mutation rate of MEN1, DAXX, ATRX, and mTOR pathway genes declined in the following order: NETs with deletion, NETs with amplification, and copy-neutral NETs. At least one LoH of tumor suppressor gene was found in all tumors with deletions, 62% of the copy-neutral, and 37% with amplifications. Furthermore, two-hit inactivation of MEN1 and DAXX was observed in NETs with deletions, at a higher frequency than in the other tumor types.

Finally, the authors identified three distinct groups with significantly different relapse-free survival: insulinomas, non-functional copy-neutral, and non-functional Amp/Del. Of these groups, the non-functional Amp/Del exhibited the least favorable clinical outcome, while insulinomas exhibited the longest relapse-free survival. Importantly, the presence of mutations in DAXX and ATRX in NETs with deletions/amplifications demonstrated shorter RFS only in the first two-year period during the study.

On the basis of chromosomal aberrations, Scarpa et al. [30] classified G1/G2/G3 NET samples into four groups: (1) recurrent pattern of whole chromosomal loss (RPCL); (2) limited copy-number events, many of which were losses affecting chromosome 11; (3) polyploidy with the highest somatic mutation rate; and (4) aneuploidy. Nevertheless, the impact of these groups on outcome prediction has not been evaluated.

The recurrent pattern of the whole chromosomal loss (RPCL) phenotype was significantly associated with alternative lengthening of telomere activation (positive ALT status) and ATRX/DAXX-mutant specimens exhibited ALT. ATRX/DAXX mutations were found to be significantly correlated with mutations in mTOR pathway genes, with patients harboring these mutations exhibiting poor prognosis. In light of these findings, the RPCL subtype can be considered to align with the previously mentioned MEN1/ATRX/DAXX alpha-like subtype [64], the second group delineated by Lawrence et al. [38], and the non-functional Amp/Del subtype suggested by Hong et al. [57]. Another group (limited copy-number events, many of which were losses affecting chromosome 11) is consistent with the NF-Neutral and Ins-Neutral groups suggested by Hong et al. [57], and the second group suggested by Lawrence et al. [38].

The studies addressing the NET genome clearly indicate the existence of a group of non-functioning NETs enriched in MEN1/DAXX/ATRX mutations, structural aberrations, mTOR pathway alterations, positive ALT status, and unfavorable prognosis. The clinically-advantageous group is also non-functional, has MEN1 mutations, chromosome 11 LoH, and limited copy-number events. Both groups exhibit alpha-cell gene expression signature (for instance, ARX) and PDX1 repression. Insulinomas constitute a distinct, relatively benign group, exhibiting beta cell-like transcriptomic profile and either YY1 mutations coupled with limited copy-number events, or amplification events in chromosomes 3, 5, 7, and 13. The poor prognosis associated with tumors harboring mutations in MEN1/ATRX/DAXX genes is at odds with the findings of certain studies [32,65]. However, it is important to consider the composition of the cohorts: MEN1/DAXX/ATRX-mutant NETs demonstrate higher risks and shorter survival compared to insulinomas, or to cohorts with grade 1 or 2 tumors, and a generally low number of metastatic and deceased patients. In the study by Park et al. [65], the patients with MEN1/DAXX/ATRX protein inactivation exhibited longer overall survival in the metastatic group; however, approximately one-fifth of patients died during the follow-up period, and 10% had recurrences after the curative resection. These observations highlight the existence of another rare group of non-functional tumors with a higher relapse risk and poorer overall survival, which requires further investigation.

## 3. Transcriptome/Proteome-Based Classification

Genetic alteration may not always result in concordant RNA or protein expression changes. Regulation of multiple molecular pathways in tissue is complex, and the impact of single nucleotide mutations or large chromosomal aberrations on cellular processes might be hard to predict. Consequently, the next step in investigating the molecular characteristics of heterogeneous neuroendocrine tumors is bulk RNA sequencing (RNA-seq) and proteomics. These technologies provide scientists with the opportunity to characterize bulk profiles of whole samples. A number of studies have employed these approaches, although their use has often been limited to characterizing predefined genetic subtypes, identifying correlations between mutations and gene expression, and testing for known islet markers. To the best of our knowledge, only two studies have aimed to identify transcriptomic subtypes of NETs, and their results, in part, agree with genetic investigations.

Microarray mRNA and microRNA expression data were used by Sadanandam et al. [66] to discover subtypes of pancreatic neuroendocrine tumors. The consensus clustering-based non-negative matrix factorization (NMF) method was employed for the analysis, which revealed three subtypes. These subtypes were named, in accordance with the clinical data, as insulinoma/islet (IT), metastasis-like primary (MLP) and intermediate tumors.

Primary tumors from the MLP subtype were found to cluster together with liver and lymph node metastasis. The majority of MLP samples were associated with metastasis and did not display any specific hormonal hypersecretion. However, three functional specimens belonged to the MLP group, indicating that a small part of functional cases might be aggressive. The remaining subtypes exhibited distinct hormonal activity, with IT being enriched with insulinomas and intermediate tumors being completely nonfunctional.

The authors demonstrated that approximately half of the intermediate subtype samples had MEN1 mutation, while only 12 and 12.5% of IT and MLP samples, respectively, were MEN1-mutant. DAXX/ATRX mutations were distributed as follows: 42% of intermediate samples, 28% of MLP samples, and none of the IT samples. Notably, five out of eight MEN1/DAXX or MEN1/ATRX cases belonged to the intermediate subtype. The authors found that mutations in mTOR pathway genes (PTEN, TSC2) were not significantly associated with any of the NET subtypes. Consequently, the authors dichotomized non-functioning specimens into two transcriptomic subtypes, with MLP having an unfavorable prognosis. The following observations support the assumption that MEN1/DAXX/ATRX-mutant tumors do not represent the most disadvantageous group.

In light of the limitations of the current WHO classification, the authors sought to compare their transcriptome-based subtypes and WHO-defined histopathological classes. The distribution of G1 samples was as follows: 59% belonged to the intermediate subtype, 28% to the IT subtype, and the remaining 13% to the MLP subtype. G2 samples exhibited the following distribution: 56%—MLP subtype, 33%—MEN1 subtype, 11%—IT subtype. All four G3 samples fell within the MLP subtype. This data indicate that the morphological characteristics of samples do not directly correspond to their molecular features.

The authors propose that the MLP subtype expresses stem cell-related genes and originates from immature beta cells. At the same time, the IT subtype originates from adult beta cells. However, this conclusion should be considered with caution, as the knowledge of embryonic human pancreas development at that time was incomplete and relied mainly on murine data. Additionally, a number of studies have demonstrated that a subset of non-functioning tumors demonstrated similarity to alpha cells. Therefore, the comparison performed only on the basis of beta-cell gene expression might lead to inaccurate conclusions. Moreover, the inclusion of metastatic lesions in expression analysis may impact the accuracy of the analysis, and calls into question the results regarding the transcriptomic landscape of the MLP subtype. Conversely, Chan et al. [64] observed no expression or methylation differences between primary tumors and metastases when using PCA or unsupervised hierarchical clustering.

Notably, Chan et al. [64] demonstrated that the ATRX/DAXX/MEN1-mutant and wild-type NETs from Sadanandam et al. [66] have significant positive and negative correlations with their ATRX/DAXX/MEN1-mutant expression signature, respectively. Using gene-set enrichment analysis, the authors have demonstrated the enrichment of alpha cell signatures only in the ATRX/DAXX/MEN1-mutant tumors from the Sadanandam et al. [66] data set.

In contrast to Sadanandam et al. [66], Yang et al. [67] were able to identify the alpha cell-like group in gene and protein expression data. The cohort of 84 patients included both neuroendocrine tumors (78 samples) and carcinomas (6 samples). The results of the analysis suggest the existence of four molecular subtypes, namely PDX1-high, Alpha cell-like, Stromal/Mesenchymal, and Proliferative, each with distinct genomic abnormalities, enriched pathways, and histo-pathological parameters. Functional status showed no significant association with any of the subtypes, although 5 out of 10 specimens with endocrine hyperfunction were in the PDX1-high cluster. To draw more confident conclusions regarding the association of functional status with the suggested subtypes, it is necessary to increase the cohort size. In contrast to Sadanadam et al., none of the identified subtypes was characterized by significantly higher rates of metastatic dissemination.

The Proliferative subtype exhibited enrichment of cell cycle-related gene sets and protein sets involved in DNA replication and transcription. As expected, the samples demonstrated a significantly reduced overall survival probability. Furthermore, the subtype demonstrated unique genetic characteristics, encompassing samples with TP53, BRCA2, CTNNB1, DAXX, CUX1, KDR, SETD2, and CHEK2 mutations. Notably, this subtype included five out of six carcinoma specimens, as well as one G1 and one G2 sample.

The Alpha cell-like subtype displayed transcriptomic similarity to pancreatic alpha cells, and featured significantly higher expression of the ARX transcription factor compared to normal islets and other subtypes. G1 and G2 NETs constituted equal proportions within the group. The samples exhibited high levels of arginine, glutamate, and glutamine metabolic enzymes, as well as high mitochondrial-protein content and oxidative phosphorylation genes. Consistent with previous findings, the majority of MEN1, DAXX and ATRX-mutant specimens belonged to the Alpha cell-like subtype.

One more subtype, PDX1-high, exhibited an enrichment in transcription factors, characteristic of human multipotent pancreatic progenitors (ONECUT1/2, PTF1A, SOX9) and endocrine precursor cells (NEUROG3). Notably, this subtype demonstrated the most favorable prognosis.

The transcriptomic/proteomic landscape of the remaining Stromal/Mesenchymal subtype demonstrated involvement of stromal (gap junction, focal adhesion pathways, and proteins associated with extracellular matrix) and immune (immune responses) microenvironment, as well as epithelial–mesenchymal transition. Despite the significant enrichment in mesenchymal and endothelial gene sets and, as a consequence, the likely larger proportion of these cell types in tumor samples, IHC results showed no increase in microvessel density. Furthermore, all four Von Hippel–Lindau patients fell within the subtype, thereby leading to this subtype being the only one enriched in hypoxia pathway genes. Through the use of a wide range of methodologies, the authors have demonstrated the activation of the Hippo signaling pathway in the Stromal/Mesenchymal subtype. The number of grade 1 and grade 2 tumors was identical within the subtype, and one NET was G3.

Despite the apparent similarity in transcriptomic/proteomic study designs (cohort composition and clustering algorithm), the described results showed weak consistency. The subtypes suggested by Yang et al. [67] exhibited more distinct molecular features, and overlapped with some genomic and epigenomic groups. At the same time, the subtypes suggested by Sadanandam et al. [66] showed significant differences in survival and metastatic potential. Nevertheless, at least two groups of pancreatic NETs appeared to be robust: alpha cell-like ARX-high (MEN1-mutant) and beta cell-like PDX1-high. The relationship between the proliferative group [67] and the metastasis-like group [66] is not immediately apparent. However, the existence of an unfavorable, predominantly high-proliferating and hormonally inactive group is evident.

## 4. Epigenome-Based Groups

The epigenome is a set of chemical modifications that regulate gene expression without changing the DNA sequence itself. Epigenomic modifications activate or silence transcription through multiple mechanisms, including enhancer and silencer methylation, promoter and gene-body methylation, and covalent histone modifications. As the occurrence and growth of neoplastic cells might be due to both genetic and epigenetic alterations, it is crucial to investigate cancer tissue using both approaches. Similar mutation patterns may result in different molecular alterations if occurring in different cell types. It is important to note that relying on genome analysis alone may be misleading. Epigenetic modifications are considered to reflect the lineage of origin, as well as acquired dynamic changes, thereby representing a new resolution in tumor analysis. As the increasing number of studies emphasize the pivotal role of methylation profiling in investigating the mechanisms of neoplastic growth, more research in specialized fields of oncology is conducted. A classification system based on DNA methylation data was proposed for central nervous system tumors [68,69] and sarcomas [70], and demonstrated its application in a routine diagnostic setting [68].

A number of studies have been conducted with the objective of elucidating the epigenetic patterns of pancreatic NENs. Cejas et al. [71] identified three distinct subtypes of pancreatic non-functional NENs (A-, B- and C-types), based on the results of chromatin immunoprecipitation sequencing (Chip-seq) profiling of H3K27ac- and H3K4me2-marked enhancers. Tumors of the A- and B-types exhibited robust signals in enhancers with sites over the ARX or PDX1 gene, respectively, which appeared to be consistent with the differential expression results. It is noteworthy that all three subtypes included MEN1-mutant samples, although most of them fell within the A-type.

Subsequently, an additional cohort of pancreatic NETs was analyzed, revealing consistency between IHC staining and tissue microarray, RNA-seq and FiT-seq (fixed-tissue chromatin immunoprecipitation sequencing) subtype labels, thereby indicating that the results of the analysis may be implemented in routine clinical practice.

The IHC staining revealed PDX1-positive, ARX-positive, double-positive and double-negative samples. It is noteworthy that all insulinomas were PDX1-positive and, thus, belonged to B-type NETs. However, there appeared to be an uncertainty in classifying double-negative cases. Such tumors might occur as a consequence of technical failure of IHC staining, correspond to the A-type with low ARX expression, or correspond to the C-type. Interestingly, all subtypes expressed higher levels of mature beta- or alpha-cell genes, rather than progenitor-specific genes.

Furthermore, the authors demonstrated that the prognostic utility of ALT status in combination with subtype label was superior. In a cohort of 103 NEN patients, relapses occurred in every ARX+ALT+ tumor, in only 9% of ARX+ALT− tumors, and in a single PDX1+ALT− tumor. The results are in accordance with the aforementioned findings by Yang et al. [67] and with the established indolence of insulinomas (mainly PDX1+), although the group with a more favorable clinical outcome has been expanded to encompass all PDX1+ cases, irrespective of functional status.

It is worth noting that in another study [72], three metastatic insulinomas were included and exclusively displayed positivity for ARX (>10% of cells) and ALT, indicating that PDX1 and ARX expression are merely partially bound to hormonal activity.

The results also match with those of Chan et al. [64], where the PDX1 gene was shown to have a hypermethylated promoter and was repressed in alpha-like MEN1/ATRX/DAXX-mutant tumors.

Similar results were obtained by Boons et al. [73], although subtyping was based on PDX1 gene methylation alone and revealed two groups, also referred to as A and B. The former group might comprise A- and C-type tumors, while the latter comprise B-type samples from the previous work [71]. Besides known genetic features (ATRX/DAXX/MEN1 mutations, high number of copy-number events in A group) and risk of recurrence, the authors examined the association between subtype and hormone hypersecretion, grade, presence of distant metastasis and lymphovascular invasion. Distant metastases were more frequent in the A group, while grade and presence of lymphovascular invasion showed no significant differences. Due to the limited number of functional cases, only insulinomas were significantly enriched in the B group (six out of eight samples).

For the B group chromosome 11 and 13 losses and chromosome 5, 6, 7, 8, 9, 13, and 21 gains were common. That partially matches results for insulinomas from Hong et al. [57].

Although the diversity of pancreatic NETs seems to be far more complex, authors utilized only one parameter to dichotomize samples, and augmented the knowledge about the PDX1-high (B type) subtype.

In another study, Domenico et al. [74] employed DNA-methylation analysis to investigate the plausible origin of G1 and G2 neuroendocrine tumors and clarify the classification from Cejas et al. [71]. The research initially determined three phyloepigenetic subtypes, according to the similarity of the methylation landscape of 125 samples to that of normal pancreatic alpha- and beta-cells. The beta-like subtype clearly segregated from the other two subtypes, and demonstrated scarcity of MEN1/ATRX/DAXX mutations, few copy-number events and low tumor stage. The alpha-like subtype was enriched with MEN1-mutant specimens and few copy-number alterations, and consisted of T1 or T2 tumors. The majority of intermediate subtype samples were T3 or T4, G2 grade, harbored MEN1 and/or ATRX/DAXX mutations and showed increase in copy-number events.

The results of the staining strengthened the association of PDX1 and ARX status and molecular classification. All but one beta-like tumors expressed PDX1, and alpha-like tumors were ARX-positive. A total of 86% of intermediate NETs were positive for ARX, and the only four double-negative specimens fell within the subtype. In spite of high concordance with immunostaining, novel classification (beta-like, alpha-like, and intermediate) displayed a more accurate prediction of the risk of relapse, in comparison to PDX1/ARX subtyping.

Although the existence of the PDX1 (beta-like) subtype was proved, contrary to the previous studies [71] authors revealed the heterogeneity in ARX-positive cases with significantly different outcomes. The subsequent division of the intermediate subtype into ATRX/DAXX/MEN1-mutant and ATRX/DAXX/MEN1 wild-type groups upon consensus clustering did not advance the statistical significance of relapse prediction; however, the mutant group showed closer epigenetic similarity to the alpha-like group, and most of its specimens were ARX-positive. Thus, the data suggest a transition from ARX+ alpha-like (MEN1-mutant) to ARX+ intermediate (ATRX/DAXX/MEN1-mutant) type and from beta-like to intermediate wild-type tumors.

The intermediate-mutant and alpha-like tumors might strongly overlap with the second group from Lawrence et al. [38] and the A-type from Cejas et al. [71] and Boons et al. [73], whereas the beta-like might overlap with the B-type from Cejas et al. [71] and Boons et al. [73].

In the study by Lakis et al. [54], unsupervised clustering revealed three methylation subtypes. The first (T1) subtype demonstrated the similarity to the PDX1-high, or beta-like subtype, with high expression of PDX1 due to hypermethylated CpG sites in the gene. T1 specimens were mainly wild-type for ATRX/DAXX/MEN1, comprising the majority of functional tumors, and showing higher expression of markers associated with beta cells. The T2 and T3 subgroups displayed features of the aforementioned intermediate and alpha-like epigenetic subtypes, respectively, with enrichment of MEN1 mutations and high ARX expression. Of note, no differences in the methylation levels of the ARX gene was observed between the subtypes, suggesting that, in this case, methylation might not be the cause of altered gene expression. Tumors of the T2 subtype frequently harbored DAXX/ATRX mutations, as well as carrying 80% of mutations in the mTOR pathway genes, had a higher tumor mutation burden and longer telomeres, and presented a high frequency of alternative lengthening of telomeres. Moreover, subtype T2 was enriched for tumors with recurrent LOH in chromosomes 1, 2, 3, 6, 8, 10, 11, 15, 16, 21 and 22, and the T3 subtype had mainly diploid genomes with recurrent loss of chromosome 11.

The research doubts the T1 (PDX1) subtype being the one with the most favorable prognosis, as the T3 subtype consisted of specimens of lowest grade and stage, although no data on metastasis or recurrence status were presented. Interestingly, all 2 G3 tumor specimens belonged to the T1 subtype.

Methylation analysis also supported the potential benefit from alkylating agent temozolomide in the intermediate subtype. Response to this drug is associated with deficiency of DNA repair enzyme MGMT in patients with NETs [75,76,77]. Gene-body methylation of MGMT was significantly lower in tumors in the subgroup T2 compared to tumors in the other groups (T1 and T3). T2 tumors also harbored recurrent loss of chromosome 10q arm, including the loci of MGMT. Low methylation in the gene body, together with LOH, could potentially drive low expression of MGMT within the T2 subgroup. Notably, the same pattern was observed in the first group in Lawrence et al. [38]. Thus, authors supplemented the characterization of epigenetic subtypes with essential data on genetic alterations and mRNA expression, which were demonstrated to be consistent with the data from previous works.

In spite of promising findings with respect to the beta-like and alpha-like groups for the understanding of pancreatic NETs cell of source, the PDX1 and ARX staining were observed not to independently correlate with recurrence-free survival in a large cohort of non-functional NETs [78].

## 5. A Promising Approach: Single-Cell Transcriptomics

As was shown above, there is still a subset of samples (without PDX1 and ARX expression and MEN1/DAXX/ATRX wild-type), which do not match precisely with any genomic/epigenomic subtype and which may show involvement of the microenvironment in molecular processes. Analyzing the immune and stromal microenvironment may assist in clinical discoveries and at least narrow down the list of potential actionable druggable targets. Furthermore, bulk RNA-seq based classification is limited to profiling the averaged tumor transcriptome, while the presence of different shares of distinct cellular populations may form it.

A prospective approach to profiling each cell and analyzing components of the tumor and surrounding tissue is single-cell RNA-seq. Although it often fails to detect low-abundant transcripts and the expression data, the same as in bulk RNA-seq and proteomics, it is subjected to batch effects; current tools manage to diminish their influence and discover important tumor biology. To the best of our knowledge, there are only two published works focusing on NETs at single-cell resolution.

Yu Zhou et al. [79] explored five samples (from primary and metastatic lesions, peripheral blood and normal liver) from one patient with grade 2 NET in the pancreas. The authors demonstrated intratumor functional heterogeneity in the epithelial compartment and observed different expression programs in clusters of malignant cells in primary and metastatic lesions, highlighting the pivotal role of a single-cell approach in NETs. Moreover, one cluster of malignant cells was marked with expression of genes associated with proliferation (CDK and CDKN1 family genes and E2F family members) and drug resistance or sensitivity (DHFR, TOP2A and BCL2). A couple of subtypes were associated with immune response, one with immune-checkpoint activation and the other one with inhibition. One of the subtypes appeared to be dominant in the primary tumor lesions. Gene expression-based trajectories revealed plausible changes acquired during tumor progression. The authors characterized subtypes of immune cells and cancer-associated fibroblasts and built a signature to predict recurrence—simultaneous protein expression of PCSK1 and SMOC1 was associated with higher metastasis risk.

In a recent study [80], Hoffman et al. analyzed primary tumors from four patients. Differential gene expression and regulatory network analysis in malignant cells supported the lineage plasticity in pancreatic NETs. The authors confirmed the observed earlier inefficiency of classical immune checkpoint blockade due to little-to-no expression of its targets in tumors (PDL1, PDL2, HLA-G, LGALS9, VSIG and VSIR). Only CTLA4 and TIGIT exhibited high expression in regulatory T cells. Conversely, myeloid cells were marked with frequent and high expression of immunosuppressive genes. Thereby, given the known failure of conventional immunotherapy in preclinical trials, the authors suggest targeting non-classical pathways and investigating alternative ways to influence tumor growth (for instance, using anti-VISTA immunotherapy).

The authors of these two papers did not aim to examine tumors for transcriptomic similarity to the molecular subtypes described above. Hence, they did not link bulk profile with intratumor heterogeneity, although the results they obtained are of significant value.

A major limitation of existing scRNA-seq studies is small cohort size: the results cannot be generalized to all cases. However, scRNA-seq has a potential to identify molecular subtypes of NETs, taking tissue heterogeneity and microenvironment contribution into account if applied to larger cohorts [81,82]. The heterogeneity of mechanisms of tumor evolution and the hypothesis of CNV gain during transition to more aggressive stages or metastasis encourage the study of CNV clonal dynamics, which can also be investigated using scRNA-seq data from tumor tissue [83]. In the light of the existing hypothesis of some NENs resembling pancreatic progenitor cells, it is of particular importance to investigate the similarity of samples to embryonic cells. Although a couple of studies showed progenitors’ expression patterns in NETs [66,67], the problem requires deeper analysis using integrating embryonic scRNA-seq data and comparison with different intermediate progenitor populations [84,85,86].

## 6. Conclusions

Neuroendocrine tumors of pancreatic origin are challenging and miscellaneous neoplasms. They are poorly studied, although the understanding of their molecular landscape has improved profoundly in the past 10 years. Despite the worldwide recognized grade-based classification, tumors within slowly proliferating groups have shown an unexpected range of outcomes. This urges the need for alternative modes of investigation. The crucial issue in pancreatic neuroendocrine tumors is to distinguish aggressive, metastatic specimens from the relatively indolent, and to detect groups of patients for closer postoperative monitoring. Genetic and molecular studies have augmented the knowledge of pathogenesis and potential targetable cellular processes. We observed the considerable consistency between subtypes described in different studies. This has become attainable by the multimodal approach of DNA/RNA sequencing, methylation profiling, and histological examination to characterize tumors.

Three groups of tumors were described independently, by several groups (Figure 1, Table 1). PDX1-high tumors mainly comprised insulinomas and showed high similarity to pancreatic beta cells. They are determined by high levels of PDX1 expression, absence of DAXX/ATRX pathogenic variants, and ALT-negative status. Alpha-like tumors, which are often non-functional, although expressing pancreatic alpha cell-specific genes (ARX, IRX2), can be divided into two distinct groups with different prognoses. Unbeneficial intermediate alpha-like specimens show frequent MEN1, DAXX/ATRX, mTOR genes’ alterations, positive ALT, and frequent amplification and deletion events. The remaining less-aggressive group has frequent MEN1 mutations, negative ALT and, in contrast to intermediate tumors, more often expresses GCG and is characterized by limited copy-number alteration events. In addition, the existence of another, the most disadvantageous group, may be possible. Its distinctive features are high-grade and increased proliferation. Other tumors, that did not fall into any group showed variable clinical outcome and diverse molecular characteristics: gains of one or more chromosomes or variable patterns of aneuploidy, absence of MEN1/DAXX/ATRX mutations, negative ALT status, and double positivity for PDX1/ARX. Tumors with mutations in VHL likely represent a distinct group of pancreatic NETs. Tumors which were not assigned to any group still make up a significant share of samples, and require further investigation.

In further investigation or clinical application, high diagnostic precision is unlikely to be reached using single methodology-based approaches, because the groups share several defining features. Precision can be feasible only through integration of various omics data. Although the studies outlined in this review have brought greater clarity to the biology of tumors, the prediction of metastasis and response to definite treatment requires more specimens to reach significance.

## Figures and Tables

**Figure 1 ijms-25-11017-f001:**
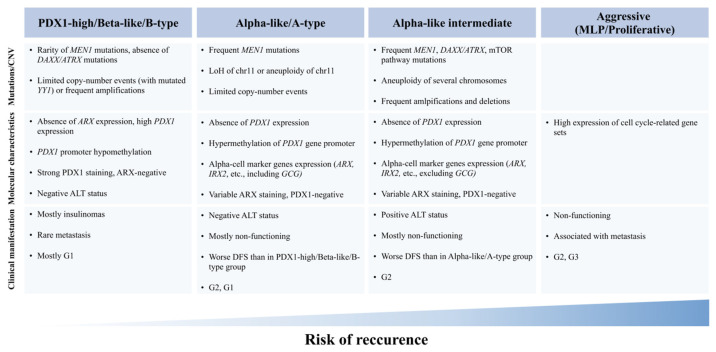
Summary of tumor characteristics shared by most specimens within the groups.

**Table 1 ijms-25-11017-t001:** Description of tumor groups.

	Article	Primary Data for Classification	Other Methods	Subtype (Proportion)	Genome	Transcriptome/ Proteome	Epigenome	Staining	ALT Status	Prognosis	Hormonal Activity	Morphology Grade	Metastasis	Summary Subtype
**Genome**	[38]	deep hybridization captures DNA sequencing of 637 genes, copy-number groups	total RNA and mRNA sequencing, methylation microarray, low coverage whole-genome sequencing (WGS)	Group 1 (26%)	a recurrent LoH affecting chromosomes 1, 2, 3, 6, 8, 10, 11, 16, 21, and 22; frequent MEN1, ATRX, DAXX mutations; PTEN, TP53 mutations	low MGMT expression	-	-	-	generally less favorable outcomes, contains the only three patients who progressed during the study	-	G1, G2, G3	4/10 metastasized	Alpha-like intermediate
				Group 2 (41%)	MEN1 mutation and chromosome 11 LoH	GCG expression	-	-	-	relatively favorable pathological and clinical outcomes	-	G1, G2	none	Alpha-like
				Group 3 (33%)	variable patterns of aneuploidy	-	-	-	-	variable clinical outcomes	-	G1, G2	3/13 metastasized	
	[64]	Sanger sequencing to genotype the ATRX, DAXX, and MEN1 genes	RNA sequencing, DNA methylation, IHC on H3K4me3, H3K9me3, H3K27me3, H3K36me3, APOH	A-D-M-mutant (58%)	ATRX/DAXX/MEN1-mutant	high ARX and low PDX1 expression, high expression of “liver-specific” genes (APOH, ALDH1A1, FGB, APOC3, HNF1A, HNF4A), complement and coagulation pathway genes (SERPINA1, FGA, F10, CP, MT3), retinol metabolism, and drug metabolism; signature related to that of alpha cells (higher ARX, IRX2 and TM4SF4), but lower GCG	PDX1 promoter hyper-methylation, APOH, CCL15, EMID2, PDZK1, HAO1, BAIAP2L2, and NPC1L1 hypomethylation	APOH positive staining in 70 ± 2.5%	-	a worse recurrence-free survival, larger tumor size, higher tumor stage	-	-	8/18 (lymph node), 15/19 (distant metastasis)	Alpha-like or Alpha-like intermediate
				A-D-M-wild-type (32%)	ATRX/DAXX/MEN1-wild-type	TACR3 high expression	TACR3 hypomethylation	APOH positive staining in 18 ± 2.0%	-		-	-	4/14 (lymph node), 4/14 (distant metastasis)	
	[57]	clinical status and copy number variations	WGS/WES, RNA sequencing, IHC	Ins-Neutral (14%)	limited copy-number events, YY1 T372R mutations	-	-	higher MTOR activity (p-S6 IHC score)	-	nearly no relapse events	insulinomas	G1, G2	-	PDX1-high
				Ins-Amp (26%)	frequent amplifications, specifically in chromosomes 7, 3 p, 5q, 13q, TSC1, TSC2 amplifications	-	-	-	-	nearly no relapse events	insulinomas	G1, G2	-	PDX1-high
				NF-Amp (18%)	frequent amplifications, specifically in chromosome 4 and 14q, mutations in MEN1, DAXX, ATRX, mTOR-related genes	-	-	-	-	the worst relapse-free survival	non-functional	G1, G2, G3	-	Alpha-like intermediate
				NF-Del (16%)	frequent deletions, highest tumor-mutation burden, mutations in MEN1, DAXX, ATRX, mTOR-related genes, inactivation of tumor suppressor genes	-	-	-	-		non-functional	G1, G2	-	Alpha-like intermediate
				NF-Neutral (26%)	limited copy-number events	-	-	-	-	relapse-free survival better than in NF-Amp/Del	non-functional	G1, G2, G3	-	Alpha-like
	[30]	high-density single nucleotide polymorphism (SNP) arrays, arm-length copy-number patterns	WGS, C-tailing qPCR (for ALT), RNA sequencing	recurrent pattern of whole chromosomal loss (RPCL) (29%)	recurrent pattern of whole chromosomal loss affecting chromosomes 1, 2, 3, 6, 8, 10, 11, 15, 16 and 22	-	-	-	ALT-positive	-	-	mostly G2	-	Alpha-like intermediate
				limited copy-number events, many of which were losses affecting chromosome 11 (39%)	losses of chromosome 11	-	-	-		-	-		-	Alpha-like
				polyploidy (7%)	gain of all chromosomes, the highest somatic mutation rate	-	-	-		-	-		-	
				aneuploidy (25%)	whole chromosome gains affecting multiple chromosomes	-	-	-		-	-		-	
**Transcriptome/proteome**	[66]	mRNA and miRNA sequencing	targeted next-generation DNA sequencing, IHC, RT-PCR	islet/insulinoma tumors (25%)	rare MEN1, no DAXX/ATRX mutations	high expression of insulinoma-associated genes (INS, IAPP, INSM1), high expression of mature β-cell–specific genes and genes involved in central carbon metabolism (PDX1, INS1, GCK, SLC2A2, G6PC2, PC, ME1)	-	-	-	-	insulinomas	mostly G1	none	PDX1-high
				metastasis-like primary (38%)	rare MEN1, one third ATRX/DAXX mutations	high expression of genes associated with fibroblasts/stroma, stem cells, and hypoxia, expression of pancreatic progenitor-specific genes (HNF1B, GATA6, LIN28B)	-	-	-	-	non-functional	G1, G2, G3	associated with metastasis	Aggressive
				intermediate/MEN1-like (37%)	half with MEN1/DAXX/ATRX mutations		-	-	-	-	non-functional	G1, G2	16% with metastasis	Alpha-like or Alpha-like intermediate
	[67]	transcriptomic and proteomic profiling	WES, IHC	PDX1-high (24%)		high PDX1 expression, high ONECUT1/2, PTF1A, SOX9 and NEUROG3 (transcription factors expressed in pancreatic progenitor cells) expression, activation of HRAS and NRAS, inhibition of NF1	-	-	-		5 functional, 10 non-functional	mostly G1	9/19 metastasized	PDX1-high
				Alpha cell-like (33%)	MEN1, DAXX mutations, the only with TSC1/2 mutations	high ARX expression, similarity to α-cells, genes involved in oxidative phosphorylation and abundance of mitochondrial proteins, significantly higher protein abundance of arginine (ARG2) and glutamine/glutamate metabolic enzymes (GLS, GLUL and GLUD2), inhibition of MEN1, DAXX and ATRX	-	-	-		3 functional, 11 non-functional	G1, G2	15/28 metastasized	Alpha-like or Alpha-like intermediate
				Stromal/Mesenchymal (30%)	VHL mutations	YAP1/WWTR1(TAZ) activation, genes involved in epithelial-to-mesenchymal transition and immune responses, activation of focal adhesion and gap junction pathways, increased abundance of proteins associated with the extracellular matrix, significantly higher transcriptomic similarities to mesenchymal and endothelial cells, activation of KRAS and inhibition of TSC2	-	-	-		1 functional, 11 non-functional	G1, G2, 1 NEC	10/25 metastasized	
				Proliferative (13%)		enrichment of cell cycle-related gene sets (E2F targets and G2M checkpoints), increased DNA replication and transcription	-	-	-	associated with a reduced overall survival probability	1 functional, 7 non-functional	G1, G2, G3, 5 NECs	7/11 metastasized	Aggressive
**Epigenome**	[71]	Chromatin immunoprecipitation sequencing (ChIP-seq)-derived profiles of H3K27ac-marked candidate enhancers	ChIP-seq for H3K4me2, RNA sequencing, IHC, FiT-seq, telomere-specific FISH (for ALT status)	A type (50%)	-	variable ARX expression, α-cell-specific transcripts	high H3K27ac signal in ARX and IRX2 (α-cell-specific loci), low or absent H3K27ac signal at PDX1, enhancers enriched for α-cell-specific sites (areas of chromatin selectively open in normal α-cells)	variable ARX, PDX1-negative or ARX/PDX1-double negative	half ALT-positive	Worst disease-free survival in ALT-positive tumors	non-functioinal		higher metastasis risk	Alpha-like or Alpha-like intermediate
				B type (25%)	-	high PDX1 expression, β-cell-specific transcripts	higher H3K27ac signal in PDX1 and SLC17A6, locus-wide H3K4me2 signal in PDX1 promoter, low or absent H3K27ac signal at ARX, enhancers enriched for β-cell-specific sites (areas of chromatin selectively open in normal β-cells)	high PDX1, ARX-negative			insulinomas, non-functional		lower metastasis risk	PDX1-high
				C type (25%)	-	expressed low levels of ARX and PDX1	variably marked at ARX, IRX2, PDX1 loci	ARX/PDX1-double positive			non-functional			
	[73]	methylation profile of the PDX1 region	genome-wide DNA methylation analysis	subtype A (75%)	ATRX/DAXX/MEN1 mutantions; chromosomes 1, 2, 6, 10, 11, 16, and 22 losses; chromosomes 4, 5, 7, 9, 12, 13, 14, 17, 18, 19, 20, and 21 gains	-		-	-	higher reccurence risk	non-functional	G1, G2, rare G3	21/53	Alpha-like or Alpha-like intermediate
				subtype B (25%)	ATRX/DAXX/MEN1 wild-type; chromosomes 11, 13 losses; chromosomes 5, 6, 7, 8, 9, 13, and 21 gains	-		-	-	lower reccurence risk	non-functional, insulinomas	G1, G2	2/20	PDX1-high
	[74]	genome-wide DNA methylation profiling, similarity to α/β-cells	IHC, telomeric-FISH (for ALT status)	β-like (11%)	MEN1/DAXX/ATRX wild-type, few copy-number events	-	resemble β-cells	PDX1-positive	-	the lowest risk of relapse	non-functional, insulinomas	mostly G1		PDX1-high
				α-like (15%)	MEN1 mutations, few copy-number events	-	resemble α-cells	ARX-positive	-		non-functional	G1, G2		Alpha-like
				intermediate MEN1/DAXX/ATRX-mutant (50%)	MEN1 and/or DAXX/ATRX-mutant, increasingly copy-number events	-	resemble α-cells	ARX-positive or double-negative	-	shortest disease-free survival	non-functional	mostly G2	higher risk of metastasis	Alpha-like intermediate
				intermediate MEN1/DAXX/ATRX-wild-type (24%)	MEN1/DAXX/ATRX wild-type, increasing copy-number events	-	resemble α-cells		-					Alpha-like intermediate
	[54]	DNA methylation of promoter CpG sites	C-Tailing qPCR (for ALT status), RNA sequencing, WGS	T1 (26%)	ATRX, DAXX and MEN1 wild-type, heterogeneous profiles of copy number	PDX1 higher expression	PDX1 gene hypomethylation	-		-	functional		-	PDX1-high
				T2 (43%)	ATRX, DAXX and MEN1 mutations, recurrent LOH in chromosomes 1, 2, 3, 6, 8, 10, 11, 15, 16, 21 and 22, higher tumor-mutation burden, mutations in mTOR pathway genes	lower expression of MGMT, higher ARX expression	lower methylation in the MGMT gene body, variable ARX methylation	-	ALT-positive	-	non-functional		-	Alpha-like intermediate
				T3 (31%)	mutations in MEN1, recurrent loss of chromosome 11	higher ARX expression	variable ARX methylation	-		-	non-functional	frequently G1	-	Alpha-like

## Data Availability

No new data were created or analyzed in this study. Data sharing is not applicable to this article.

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
