# Peer review of "Molecular Basis of Pancreatic Neuroendocrine Tumors"

_ijms, 2024, doi:10.3390/ijms252011017_

Round 1
Reviewer 1 Report
Comments and Suggestions for Authors
Comments
This is a well-written manuscript titled “Molecular basis of pancreatic neuroendocrine tumors”. In this study, the authors cover a comprehensive overview of genomic, transcriptomic, proteomic, and epigenomic studies of pancreatic neuroendocrine tumors (NETs), their classification, and the current understanding of their biology. The literature presented in the manuscript is of good quality and useful for readers working in the field. The figures/ tables/ schemes are appropriate and align with the text and in present form. The manuscript can be accepted after some suggested improvements. Here are the comments:
Strong Points:
- Comprehensive Overview: The passage provides a thorough introduction to pancreatic NETs, covering epidemiology, classification, and biological understanding.
- Up-to-date Information: It includes the latest WHO 2022 classification, reflecting current knowledge in the field.
- Multifaceted Approach: The text discusses the issue from various angles - clinical, histological, and molecular - providing a well-rounded view of the topic.
- Distinction between NETs and NECs: The clear differentiation between these two categories helps in understanding their distinct biological and clinical characteristics.
Limitations and Potential Criticisms:
1. Lack of Quantitative Data: While some statistics are provided (e.g., incidence rate), more quantitative data on prevalence, survival rates for different subtypes, etc., could enhance the overview.
2. Limited Discussion on Treatment: The passage focuses primarily on biology and classification, with little mention of current treatment approaches or challenges.
3. Potential Bias towards Molecular/Genetic Aspects: While the molecular and genetic aspects are well-covered, there's less emphasis on other factors like environmental or lifestyle influences on tumor development.
4. Lack of International Perspective: The epidemiological data is from the SEER database, which is U.S.-based. Including data from other regions could provide a more global perspective.
Comments on the Quality of English LanguageThis is a well-written manuscript titled “Molecular basis of pancreatic neuroendocrine tumors”. In this study, the authors cover a comprehensive overview of genomic, transcriptomic, proteomic, and epigenomic studies of pancreatic neuroendocrine tumors (NETs), their classification, and the current understanding of their biology. The literature presented in the manuscript is of good quality and useful for readers working in the field. The figures/ tables/ schemes are appropriate and align with the text and in present form. The manuscript can be accepted after some proofreading of spelling and typo mistakes.
Author Response
|
Thank you very much for taking the time to review this manuscript. Please find the detailed responses below and the corresponding revisions/corrections highlighted/in track changes in the re-submitted files. Comments 1: Lack of Quantitative Data: While some statistics are provided (e.g., incidence rate), more quantitative data on prevalence, survival rates for different subtypes, etc., could enhance the overview. |
|
Response 1: Thank you for pointing this out. We, therefore, have added quantitative data on each subtype’s prevalence in Table 1, “subtype” column. To our regret, data on survival rates for NETs’ subtypes is missing. Quantitative data on disease-free survival for each subtype is generally present only in the form of significance measure (p-value) of its differences between tumor groups. |
|
Comments 2: Limited Discussion on Treatment: The passage focuses primarily on biology and classification, with little mention of current treatment approaches or challenges. |
|
Response 2: Thank you for the comment. Current treatment approaches are enumerated in the introductory section (page 3). The multitude of challenges that occur during patient management are not in the scope of the review. To highlight the potential clinical significance of molecular/genetic classification we have added the following information on page 3 lines 107-112. “As for now, there are a limited number of ways to impact therapeutic strategy. Identifying patients with higher risk of metastasis and recurrence may influence the decision on radical resection, chemotherapy use and frequency of postoperative monitoring. In addition, defining groups of tumors with common molecular and/or genetic alterations may aid in development of novel group-specific therapies and conducting clinical trials on narrow groups of people.” Comments 3: Potential Bias towards Molecular/Genetic Aspects: While the molecular and genetic aspects are well-covered, there's less emphasis on other factors like environmental or lifestyle influences on tumor development. Response 3: Thank you for pointing this out. We have, accordingly, provided information on environmental and lifestyle influences on tumor development: pages 1-2, lines 27-31. “Pancreatic NETs’ development is assumed to be affected by several environmental and lifestyle influences. According to systematic reviews and meta-analyses, they include family history of malignancy, history of diabetes, tobacco smoking and heavy alcohol consumption [3][4].” |
Comments 4: Lack of International Perspective: The epidemiological data is from the SEER database, which is U.S.-based. Including data from other regions could provide a more global perspective.
Response 4: We agree with this comment. We have, therefore, added information about incidence of pancreatic NETs in other regions: page 1, lines 25-27.
“The incidence of pancreatic neuroendocrine tumors (NETs) is 0.48 per 100,000 persons according to the US-based SEER 18 database, 0.33 – in Iceland, 0.94 – in Norway, and 2.2 – in Japan [1].”
Reviewer 2 Report
Comments and Suggestions for Authors
The literature review “Molecular basis of pancreatic neuroendocrine tumors” by Maluchenko et al., describes four major NET subtypes with distinct molecular features (beta-like B-type, alpha-like A-type, alpha-like intermediate, aggressive), and numerous outlier subtypes. The review is comprehensive but could be better organized.
Specific Comments:
1. The Graphical Abstract tells us nothing about the four Pancreatic NET subtypes, other than the four techniques used to identify them. Either relabel the Graphical Abstract as a description of techniques, or name the four NET subtypes, and list the techniques that were used to identify them.
2. Line 656. Figure 1 “Summary of tumor characteristics shared by most specimens within the groups” would be best placed near the beginning of the review, to help organize and summarize the details to follow. Fig 1 could be included with the final introductory paragraph starting on line 94 “In this review, we summarize ...”.
3. Line 42. The assertion that there are no beta cell stem cells is from old work (Dor Y, Brown J, Martinez OI, Melton DA. Adult pancreatic beta-cells are formed by self-duplication rather than stem-cell differentiation. Nature. 2004 May 6;429(6987):41-6. doi: 10.1038/nature02520. PMID: 15129273.) Recent work by Lola Reid documents a strong case for the existence of (rare) endocrine pancreas stem cells, and should be discussed in this review.
4. Line 657. The formatting in Table 1; “Description of tumor groups”. The formatting renders Table 1 uninterpretable in the PDF file available for review.
5. Most tumors in Table 1 fall into one of the four subtypes: PDX1- high, alpha-like A type, alpha-like intermediate, aggressive. Be sure to explain clearly what distinguishes the tumor types that are not in one of the four main subtypes (e.g. Group 3, variable patterns of aneuploidy; ADM wild type; polyploidy; aneuploidy; Stromal/Mesenchymal; C type).
6. How are pancreatic NE cancers related to other NE tumors (in lung, prostate, etc.)?
7. What are the distinctions between NETs that express neuroendocrine hormones and pancreatic ductal adenocarcinoma (PDA) that express one of the neuroendocrine hormones (eg insulin, glucagon, etc.)?
8. What are the distinctions between Neuroendocrine Carcinomas (NEC) and pancreatic ductal adenocarcinoma (PDA) that express one of the neuroendocrine hormones (eg insulin, glucagon, etc.)?
Author Response
|
Thank you very much for taking the time to review this manuscript. Please find the detailed responses below and the corresponding revisions/corrections highlighted/in track changes in the re-submitted files. Comments 1: The Graphical Abstract tells us nothing about the four Pancreatic NET subtypes, other than the four techniques used to identify them. Either relabel the Graphical Abstract as a description of techniques, or name the four NET subtypes, and list the techniques that were used to identify them. Response 1: Thank you for pointing this out. We agree with this comment. Therefore, we modified the graphical abstract and added names of the four pancreatic NET subtypes, described in the review. Comments 2: Line 656. Figure 1 “Summary of tumor characteristics shared by most specimens within the groups” would be best placed near the beginning of the review, to help organize and summarize the details to follow. Fig 1 could be included with the final introductory paragraph starting on line 94 “In this review, we summarize ...” Response 2: Agree. We have, accordingly, placed Figure 1 in lines 118-119, right before the final introductory paragraph. Comments 3: Line 42. The assertion that there are no beta cell stem cells is from old work (Dor Y, Brown J, Martinez OI, Melton DA. Adult pancreatic beta-cells are formed by self-duplication rather than stem-cell differentiation. Nature. 2004 May 6;429(6987):41-6. doi: 10.1038/nature02520. PMID: 15129273.) Recent work by Lola Reid documents a strong case for the existence of (rare) endocrine pancreas stem cells, and should be discussed in this review. Response 3: Thank you for the valuable comment. We have reviewed the literature on the topic and discussed the hypothesis of the existence of endocrine pancreas stem cells in the review: page 2, lines 50-58. “There are conflicting opinions about mechanisms of self-renewal in endocrine pancreas. Mice study conducted by Dor et al. suggest that beta-cells regenerate through self-duplication, although slow [15]. In contrast, several studies showed that ducts of extrahepatic biliary trees contain stem cells and committed progenitors. Hepato-pancreatic common duct and pancreatic ducts may represent the source of pancreatic endocrine progenitors, expressing markers of early hepato-pancreatic commitment (SOX9, SOX17, PDX1), as well as markers of pancreatic endocrine maturation (NGN3 and insulin) and capable of differentiating into endocrine cells [16-18]”. Comments 4: Line 657. The formatting in Table 1; “Description of tumor groups”. The formatting renders Table 1 uninterpretable in the PDF file available for review. Response 4: Agree. The Table 1 representation was modified with minor corrections of the table’s content and format. Comments 5: Most tumors in Table 1 fall into one of the four subtypes: PDX1- high, alpha-like A type, alpha-like intermediate, aggressive. Be sure to explain clearly what distinguishes the tumor types that are not in one of the four main subtypes (e.g. Group 3, variable patterns of aneuploidy; ADM wild type; polyploidy; aneuploidy; Stromal/Mesenchymal; C type). Response 5: Thank you for pointing this out. We have added information about outlier tumors in the discussion section, page 15, lines 675-680. ”Other tumors, that did not fall into any group showed variable clinical outcome and diverse molecular characteristics: gains of one or more chromosomes or variable patterns of aneuploidy, absence of MEN1/DAXX/ATRX mutations, negative ALT status, double-positivity for PDX1/ARX. Tumors with mutations in VHL likely represent a distinct group of pancreatic NETs. Tumors, which were not assigned to any group, still make up a significant share of samples and require further investigation”. Comments 6: How are pancreatic NE cancers related to other NE tumors (in lung, prostate, etc.)? Response 6: Thank you for the question. To clarify this issue and highlight the relation between pancreatic NENs and other NE tumors in the review, we have added information about some of the similarities and differences between them in the introductory section: page 1, lines 27-28 “Pancreatic NETs are less common than those in lungs, rectum and small intestine [2]”, page 2 lines 32-34 “Like most NETs of other localizations, they express chromogranin A, synaptophysin, and neuron-specific enolase. In contrast to other types of NETs, ...”; page 2, lines 40-41 “Hormonal syndromes that aid in diagnosis are a unique feature of gastroenteropancreatic NETs”. In addition, the initial manuscript contains information on median overall survival which is estimated to be the worst in pancreatic NETs (page 2, lines 31-32). Comments 7: What are the distinctions between NETs that express neuroendocrine hormones and pancreatic ductal adenocarcinoma (PDA) that express one of the neuroendocrine hormones (eg insulin, glucagon, etc.)? Response 7: Thank you for the question. As PDAs are not in the focus of the review, they are discussed with little mention. PDA may occasionally express neuroendocrine hormones if neuroendocrine, non-malignant cells are present in the tissue. Main distinctions between NETs and PDAs include neuroendocrine cell morphology, relatively monotonous round nuclei with chromatin that has a “salt-and-pepper” appearance, positive chromogranin A, synaptophysin, and neuron-specific enolase staining, genetic alterations. Noteworthy, there are epithelial neoplasms in pancreas, that express both neuroendocrine and non-neuroendocrine markers. If such tumor has two morphologically and immunohistochemically distinct components (neuroendocrine and non-neuroendocrine, comprising > 30 % each), then it is referred to as MiNEN (mixed neuroendocrine-non-neuroendocrine neoplasm), otherwise it might represent amphicrine carcinoma or carcinoma with interspersed neuroendocrine cells without neuroendocrine morphologic features. For more information Rindi, G., Mete, O., Uccella, S. et al. Overview of the 2022 WHO Classification of Neuroendocrine Neoplasms. Endocr Pathol 33, 115–154 (2022). https://doi.org/10.1007/s12022-022-09708-2 (Section “Question 9: What Are the Morphological, Immunohistochemical, and Clinical Correlates of Mixed Neuroendocrine-Non-neuroendocrine Neoplasms in the 2022 WHO Classification of Neuroendocrine Neoplasms?”). Comments 8: What are the distinctions between Neuroendocrine Carcinomas (NEC) and pancreatic ductal adenocarcinoma (PDA) that express one of the neuroendocrine hormones (eg insulin, glucagon, etc.)? Response 8: Thank you for the question. As NECs and PDAs are not in the focus of the review, we have not discussed the distinctions between them. As mentioned above, PDAs may occasionally express hormones, but NECs can be distinguished from them primarily by neuroendocrine cell morphology, diffuse chromogranin A, synaptophysin, and neuron-specific enolase staining. We, therefore, have added information on distinctions between them, page 2, lines 75-79. “However, neuroendocrine neoplasms should not be confused with pancreatic ductal adenocarcinomas and can be distinguished from them primarily by neuroendocrine cell morphology, diffuse chromogranin A, synaptophysin, and neuron-specific enolase staining.”
|